# Joint Radar and Communications Waveform Design Based on Complementary Sequence Sets

**Haichuan Li** [1] , **Yongjun Liu** [1,*], **Guisheng Liao** [1] **and Yufeng Chen** [2]

1  National Laboratory of Radar Signal Processing, Xidian University, Xi'an 710071, China
2  Hangzhou Institute of Technology, Xidian University, Hangzhou 311200, China
*  Correspondence: yjliu@xidian.edu.cn

**Abstract:** The joint radar and communications (JRC) waveform often has a high range sidelobe, which will degrade the target detection performance of an automotive JRC system. To solve this problem, a joint radar and communications complementary waveform group (JRC-CWG) design method is proposed in this paper by exploiting the philosophy of the complementary sequence. In the JRC-CWG, the traditional unimodular communications waveforms, such as the binary phase shift keying (BPSK) waveform, are used to perform the communications function. The sum of the autocorrelation function (SACF) of JRC-CWG is optimized to minimize the sidelobe level. Furthermore, considering that the JRC-CWG has poor Doppler resilience, a Doppler-resilient joint radar and communications complementary waveform (DR-JRC-CWG) design method is proposed to improve the Doppler resilience. Finally, the simulation results show that the proposed JRC-CWG and DR-JRC-CWG have superior radar performances without the degradation in communications performance in terms of the bit error rate (BER).

**Keywords:** joint radar and communications; complementary sequence; joint radar and communications complementary waveform group; Doppler resilience

## 1. Introduction

In recent years, with the development of automotive radar and communications technologies, more and more radar and communications systems have been deployed in vehicles [1,2]. For example, in an intelligent transportation system (ITS), both radar and communications devices are integrated in a vehicle [3]. In an ITS, the vehicle needs to convey information to other cars or communications base stations via a communications device and detects targets including other vehicles, pedestrians, and roadblocks via a radar device. However, with the development of the fifth-generation (5G) communications technology and the millimeter-wave radar technology, radar and communications systems tend to use the same frequency bands, which will result in mutual interference between these two systems [4]. Since the space in a vehicle is limited, radar and communications devices have to be deployed close together, which aggravates the interference between radar and communications systems in vehicles [5]. Moreover, with the increasing demand for spectral resources for both radar and communications systems, the shortage of spectral resources becomes more and more serious [6]. To solve these problems, joint radar and communications (JRC) systems are proposed, which can alleviate the interference between radar and communications and improve the efficiency of spectral resources [7].

Usually, JRC systems can be divided into two types [8]. One type is called the co-existence JRC system, wherein radar and communications are regarded as two individuals. The co-existence JRC system aims at minimizing the mutual interference between radar and communications subsystems [9]. The other type is termed the co-use JRC system, in which the transmit waveforms are optimized to simultaneously carry out both radar and communications functions [10] to avoid mutual interference between radar and communications.

In the co-existence JRC system, radar and communications occupy different resources in some domains, such as the time domain, frequency domain [11], spatial domain [12], and so on [5]. For example, in the co-existence JRC system, radar and communications can work in different time slots [13]. The main defect of this approach is that radar and communications cannot operate at the same time. In [14,15], a spectral notching JRC system is proposed with the assumption that the bandwidth of radar is much greater than that of communications. In this system, radar and communications operate in different frequencies, although they can work at the same time. In [16], a JRC system that performs radar and communications functions in different directions is investigated. In this system, the transmit waveforms of each element of the array are optimized to synthesize specific radar or communications waveforms in the desired radar and communications directions.

To design a co-use JRC system, the main challenge is how to make an optimal trade-off between radar and communications functions [13,17,18]. In order to perform the radar function in the mainlobe of the array and convey the communications information in the sidelobe simultaneously, various methods are proposed in [19,20]. In [19], the sidelobe of the transmit beam is devised to transmit communications information at different pulse repetition intervals (PRIs), and the mainlobe of the transmit beam is designed to remain unchanged to guarantee radar performance. As an extension, the method in [19] is used for multi-user scenarios in [20]. The main shortcoming of this kind of JRC system is the low communications rate since only one communication symbol is conveyed in each PRI [19,20].

To increase the communications rate, the traditional communications waveforms are modified to perform both radar and communications functions in [21]. The key to this type of method is to modify the traditional communications waveforms to satisfy radar demands [22]. One typical example is the JRC waveform based on the orthogonal frequency division multiplexing (OFDM) waveform [23–25], which has a considerably high communications rate [26]. However, the OFDM waveform has a nonconstant modulus. In order to detect targets as far as possible, the amplifier of the radar transmitter usually operates in the nonlinear region [27]. However, this will cause serious distortion when the transmit waveform has a nonconstant modulus, and the radar and communications performance will be degraded [28]. In view of this, a JRC waveforms design method based on unimodular waveforms has been studied [29].

Nevertheless, the JRC waveforms proposed in [23–25,29] have high autocorrelation sidelobe levels. In an automotive radar, if the autocorrelation sidelobe level of the transmit waveform is high, the returns from targets with small radar cross sections (RCSs) may be submerged by the sidelobes of targets with large RCSs at receivers [30]. In view of this, to ensure the radar performance of JRC systems, transmit waveforms with low autocorrelation sidelobe levels are required in JRC systems. To design unimodular JRC waveforms with low sidelobe levels, a JRC waveform combining the linear frequency modulation (LFM) waveform and the binary phase-shift keying (BPSK) waveform is proposed in [31,32]. The JRC waveform is obtained by directly multiplying the BSPK waveform with the LFM waveform. For the BPSK in this method, the angular separation between the modulated bits "1" and "0" is set to be $\phi_\delta (\phi_\delta < \pi)$ rather than $\pi$ as with the traditional BPSK waveforms. Using this method, the designed JRC waveform in [31,32] can achieve a better autocorrelation performance with the degradation in the bit error rate (BER). In [13], based on the traditional unimodular communications waveform, the author designs a JRC waveform by optimizing the phase perturbation in each chip, which is named the optimized phase perturbation (OPP) waveform. To keep the JRC waveform with a low BER, the phase perturbation of each chip of the JRC waveform is constrained by an upper bound.

The aforementioned co-use JRC waveforms are all obtained by using single waveforms. However, the autocorrelation sidelobe of a single waveform cannot be decreased to as low as possible [33]. To further decrease the autocorrelation sidelobe level of JRC waveforms, a feasible way is to exploit the philosophy of the complementary sequence set (CSS) to design co-use JRC waveforms. A CSS contains multiple subsequences. The sum of the autocorrelation function (SACF) of subsequences in the CSS is the Kronecker delta function,

i.e., all sidelobes of the SACF of subsequences are zero [34]. This property makes CSS a promising sequence to be used to design waveforms with low range sidelobe levels, which has been shown in radar applications [35,36]. In our work, the philosophy of the CSS is exploited to design the JRC waveforms.

In this paper, based on CSSs, two co-use JRC waveforms design methods are proposed. At first, unimodular JRC waveforms with low sidelobe levels are devised, which are called the JRC complementary waveform group (JRC-CWG). The devised JRC-CWG has a considerably low sidelobe level as well as good communications performance. Each JRC-CWG contains two kinds of unimodular waveforms. One kind of waveform carry out both radar and communications functions, and are termed waveforms with communications information (WCI). The traditional communications waveforms, such as the BPSK waveforms, are employed as the WCIs. The other kind of waveform in the JRC-CWG are called the waveforms to be optimized (WTO), which are optimized to suppress the sidelobe level of the JRC-CWG. Furthermore, considering that the Doppler resilience of the designed JRC-CWG is poor [37–39], a Doppler-resilient JRC complementary waveform (DR-JRC-CWG) design method is proposed. To design the DR-JRC-CWG, the Doppler sensitivity of the designed JRC-CWG is analyzed first. Specifically, the Taylor expansion terms of the ambiguity function (AF) of the JRC-CWG near the region of zero Doppler shift are derived, and this shows that the nonzero-order Taylor expansion terms will impact the Doppler resilience of JRC-CWG. Hence, the coefficients of the low-order Taylor expansion terms are utilized in the formulated optimization problem to design the DR-JRC-CWG. To solve these optimization problems, an iterative algorithm is proposed based on the framework of the Fletcher–Reeves conjugate gradient (FR-CG) algorithm and the FFT algorithm.

The contributions of our work are summarized as follows:

(1) A new co-use JRC waveform is devised by exploiting the philosophy of the complementary sequence and is called the JRC-CWG.
(2) Compared with the JRC waveforms proposed in [31] and [13], the designed JRC-CWG has a much lower sidelobe level and better communications performance in terms of BER.
(3) A new Doppler resilient waveform design method is proposed to design the DR-JRC-CWG that is not sensitive to Doppler shift caused by the relative radial velocity between the vehicle and target.
(4) An algorithm based on the framework of the FR-CG algorithm and the FFT algorithm is proposed to design the DR-JRC-CWG.

The rest of this paper is organized as follows. In Section 2, the model of JRC-CWG and the signal processing procedure at radar and communications receivers are introduced. In Section 3, the optimization problems for the JRC-CWG and DR-JRC-CWG design are developed and the optimal JRC-CWG and DR-JRC-CWG are devised. In Section 4, several simulation results are presented. Discussions are drawn in Section 5. Finally, a conclusion is made in Section 6.

Notations: Non-bold letters and bold lower-case letters represent scalars and vectors, respectively. Bold capital letters represent matrices. The $n$-th element of vector $\mathbf{s}$ is represented as $s(n)$, and the $k$-th row and $n$-th column entry of matrix $\mathbf{F}$ is represented as $\mathbf{F}(k,n)$. $\mathbf{0}_{K \times N}$ denotes the $K \times N$ matrix of zeros. The $N$-order identity matrix is denoted by $\mathbf{I}_N$. $(\cdot)^*$, $(\cdot)^T$, and $(\cdot)^H$ stand for conjugate, transpose, and conjugate transpose, respectively. $\| \cdot \|_F$ denotes the Frobenius norm. $\odot$ represents the Hadamard product operator. $| \cdot |$ takes the absolute value of each element of a vector or matrix. $\mathrm{Re}\{\cdot\}$ denotes the real part of a complex value. $\mathbb{C}$ denotes the complex space.

## 2. JRC-CWG Model

### 2.1. Transmit JRC-CWG Model

In this paper, we consider an ITS as shown in Figure 1. In Figure 1, the yellow car that is equipped with an automotive JRC system transmits the JRC waveforms to detect the white bus and simultaneously sends communications information to it.

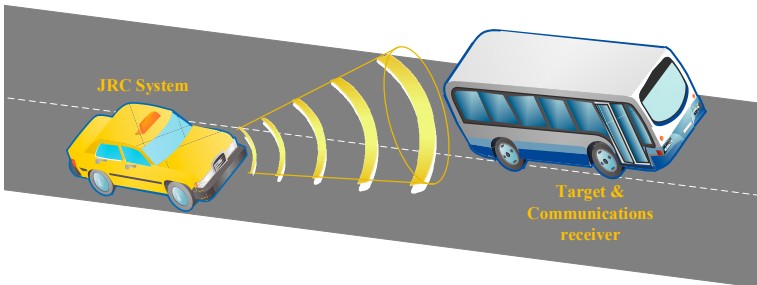

**Figure 1.** Intelligent transportation system.

In this paper, we consider that the transmit JRC waveform is the JRC-CWG, as shown in Figure 2. There are $G$ groups of JRC-CWGs transmitted in each coherent processing interval (CPI) to perform radar and communications jointly. Each JRC-CWG contains $M$ waveforms, and, hence, there are $GM$ waveforms that will successively be transmitted with a fixed PRI, denoted as $T$ in Figure 2. Let $\mathbf{S}_g = [\mathbf{s}_{g,1}, \mathbf{s}_{g,2}, \ldots, \mathbf{s}_{g,M}] \in \mathbb{C}^{N \times M}$ be the matrix formed by the waveforms of the $g$-th JRC-CWG. $\mathbf{s}_{g,m}$ is the $m$-th transmit waveform in the $g$-th JRC-CWG, which is denoted as

$$\mathbf{s}_{g,m} = [s_{g,m}(1), s_{g,m}(2), \ldots, s_{g,m}(N)]^T \tag{1}$$

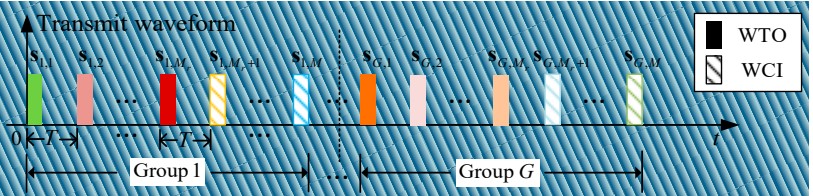

**Figure 2.** The transmit waveform model of the JRC-CWGs.

In (1), $s_{g,m}(n) = \frac{1}{\sqrt{N}} e^{j\theta_{g,m}(n)}$ denotes the $n$-th entry of $\mathbf{s}_{g,m}$; $\theta_{g,m}(n)$ represents the phase of $s_{g,m}(n)$; and $N$ is the length of $\mathbf{s}_{g,m}$.

In each transmit JRC-CWG, the first $M_r$ transmit waveforms are WTOs, which are represented by solid bars, and the last $M_c$ transmit waveforms are WCIs, which are represented by shadowed bars in Figure 2. Clearly, it can be seen that $M = M_r + M_c$. That is to say, the whole JRC-CWG consists of WTOs and WCIs. In order to perform the communications function, the traditional unimodular communications waveforms, such as BPSK waveforms, will be employed as the WCIs. The WCIs are determined by the transmitted communications information. In order to improve the radar performance, the WTOs are optimized to suppress the sidelobe level of the whole JRC-CWG.

In this paper, we consider a radar and communication system with a single transmit antenna. It is worth noting that the proposed JRC-CWG is also applicable for multiple transmit antennas and receive antennas. This will be discussed in our future work.

### 2.2. Receive Signal Model at the Radar Receiver

The receive signal model at the radar receiver is shown in Figure 3a. For the sake of simplicity, we only consider returns from one point target in this paper. It can be easily extended to more general scenarios. Furthermore, it is reasonable to assume that the velocity of the vehicles is low. Hence, the Doppler shift of the returns in one PRI can be regarded as a constant value. Thus, the sampled return from a point target at the radar receiver of the JRC system can be represented as

$$\mathbf{s}_{g,m}^r = \delta_g \rho e^{j2\pi f_d T(m-1)} \mathbf{s}_{g,m} + \mathbf{n}_{g,m} \tag{2}$$

where $\delta_g$ is the complex scattering coefficient including the path loss and RCS of the target. According to [40,41], the RCS of a target can be described using the Swerling I model, i.e., the RCS of the target does not fluctuate in each CPI, and it will only fluctuate during different CPIs. Further to this, the path loss can be regarded as a constant value in each CPI. Thus, we assume that $\delta_g$ is a constant value in the $g$-th CPI. $\rho = e^{j2\pi f_d T[(g-1)M]}$, and $f_d = 2v/\lambda$ denotes the Doppler frequency, where $v$ is the relative radial velocity between the JRC system and the target, and $\lambda$ is the wavelength of the transmit waveforms. $\mathbf{n}_{g,m} \in \mathbb{C}^{N \times 1}$ is the vector of complex Gaussian white noise.

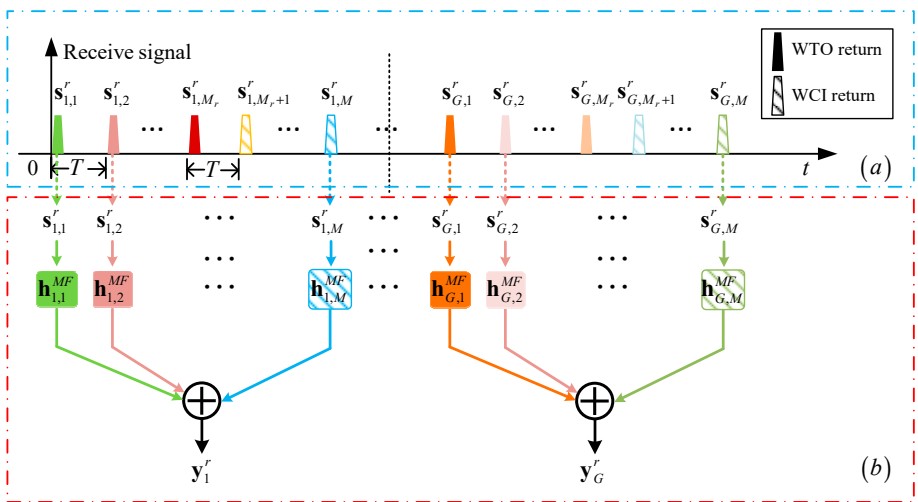

**Figure 3.** (**a**) Receive signal at the radar receiver. (**b**) Signal processing procedure.

The signal processing procedure at the radar receiver is shown in Figure 3b. $\mathbf{h}_{g,m}^{MF}$ denotes the matched filter of $\mathbf{s}_{g,m}$. As depicted in Figure 3b, the matched filter output of the $g$-th JRC-CWG returns are coherently accumulated, which can be denoted as

$$\mathbf{y}_g^r = \delta_g \rho \sum_{m=1}^{M} e^{j2\pi f_d T(m-1)} \mathbf{r}_{g,m} + \sum_{m=1}^{M} \mathbf{n}'_{g,m} \tag{3}$$

where $\mathbf{n}'_{g,m} \in \mathbb{C}^{K \times 1}$ ($K = 2N - 1$) is the matched filter output of $\mathbf{n}_{g,m}$, which is still a Gaussian white noise vector. $\mathbf{r}_{g,m} = \left[ r_{g,m}(1-N), \ldots, r_{g,m}(-1), r_{g,m}(0), r_{g,m}(1), \ldots, r_{g,m}(N-1) \right]^T$ denotes the auto correlation function of $\mathbf{s}_{g,m}$, where

$$r_{g,m}(n) = \mathbf{s}_{g,m}^H \mathbf{U}_n \mathbf{s}_{g,m}, \quad \text{for } n = 0, 1, \ldots, N-1 \tag{4}$$

and $r_{g,m}(n) = r_{g,m}^*(-n)$, for $n = 1 - N, 2 - N, \ldots, -1$. $\mathbf{U}_n$ is expressed as

$$\mathbf{U}_n = \begin{bmatrix} \mathbf{0}_{n \times (N-n)} & \mathbf{I}_{N-n} \\ \mathbf{0}_{(N-n) \times (N-n)} & \mathbf{0}_{(N-n) \times n} \end{bmatrix}, \quad n = 0, 1, \ldots, N-1 \tag{5}$$

*2.3. Receive Signal Model at the Communications User*

The signal received by the communications user is denoted as $\mathbf{s}_{g,m}^c$ in Figure 4, where [42]

$$\mathbf{s}_{g,m}^c = \delta_h \mathbf{s}_{g,m} + \mathbf{n}_{g,m}^c \tag{6}$$

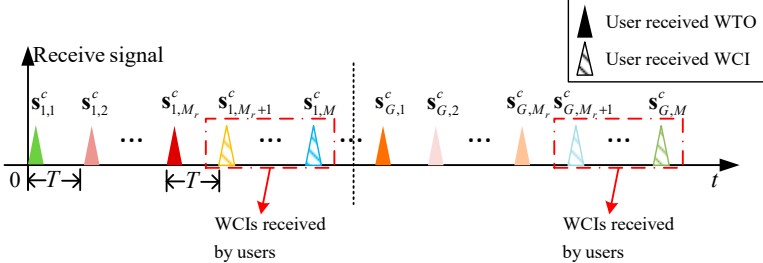

**Figure 4.** Receive signal at the communications user.

$\delta_h$ is the communications channel coefficient [42,43], and $\mathbf{n}_{g,m}^c$ is the vector of complex Gaussian white noise at the communications receiver.

Note that all the signals transmitted by the JRC systems, including the WTOs and the WCIs are received by the communications user. Only the WCIs need to be decoded by the user. It is assumed that the user knows the number of the WTOs $M_r$, and the number of WCIs $M_c$ in each JRC-CWG. In view of this, WCIs can be extracted, and the communications messages can be decoded according to the modulation scheme of the WCIs.

The communications rate and BER are usually utilized to measure the performance of communications. The communications rate of the JRC-CWG can be represented as

$$R_b = \frac{M_c}{M \cdot T} N B_0 \text{ bit/s} \tag{7}$$

where $M_c$ is the number of WCIs in each JRC-CWG; $M$ is the number of waveforms in each JRC-CWG; $T$ is the PRI; $N$ is the length of each WCI; and $B_0$ is the number of bits carried by each symbol of the WCI. For instance, if the BPSK waveforms are employed as the WCI, $B_0 = 1$. If the 16-PSK waveforms are employed as the WCI, $B_0 = 4$.

Since the traditional unimodular communications waveforms are employed as the WCIs, the BER of the JRC-CWG is the same as that of the traditional constant modulus communications waveform under the same condition.

## 3. JRC-CWG and DR-JRC-CWG Design

### 3.1. JRC-CWG Design with Low Sidelobe Level

In the ITS, when the relative radial velocity between the JRC system and the target is small or can be obtained, the effect of Doppler shift on the matched filter output can be ignored. For this case, the JRC-CWG design method is proposed without considering the Doppler shift in this section.

### 3.1.1. Problem Formulation

When the relative radial velocity $v$ is 0, (3) can be rewritten as

$$\mathbf{y}_g^r = \delta_g \mathbf{f}_g + \sum_{m=1}^{M} \mathbf{n}'_{g,m} \tag{8}$$

where $\mathbf{f}_g = \sum_{m=1}^{M} \mathbf{r}_{g,m}$ is defined as the SACF of the $g$-th JRC-CWG. To suppress the sidelobe level of $\mathbf{f}_g$ while designing the unimodular JRC-CWG, the optimization problem can be formulated as [39]

$$\begin{aligned} \min_{\mathbf{S}_g^R} \quad & \frac{1}{M} \mathbf{f}_g^H \mathbf{f}_g - 1 \\ s.t. \quad & \left| s_{g,m}(n) \right| = \frac{1}{\sqrt{N}} \quad m = 1, 2, \ldots, M, \quad n = 1, 2, \ldots, N \end{aligned} \tag{9}$$

where $\mathbf{S}_g^R = \left[ \mathbf{s}_{g,1}, \ldots, \mathbf{s}_{g,M_r} \right] \in \mathbb{C}^{N \times M_r}$ is the matrix formed by all WTOs of the $g$-th JRC-CWG. The $g$-th optimal JRC-CWG can be obtained by solving the optimization problem

in (9). To obtain $G$ optimal JRC-CWGs, $G$ optimization problems similar to (9) should be solved in parallel. The proposed optimization problem is a fourth-order polynomial minimization problem with nonlinear constraints, which is hard to be solved. Note that the objective function can be transformed into a frequency domain using the Wiener–Khinchin theorem, which can be performed by applying the FFT algorithm to reduce the computational complexity. Moreover, by only optimizing the phase of WTOs, the constraints in (9) can be cancelled. Hence, (9) can be transformed to be

$$
\min_{\boldsymbol{\Theta}_g} \left[ \sum_{m=1}^{M} \mathbf{x}_{g,m} \odot \mathbf{x}_{g,m}^* \right]^H \left[ \sum_{m=1}^{M} \mathbf{x}_{g,m} \odot \mathbf{x}_{g,m}^* \right]
\tag{10}
$$

where $\boldsymbol{\Theta}_g = \left[ \boldsymbol{\theta}_{g,1}, \ldots, \boldsymbol{\theta}_{g,M_r} \right] \in \mathbb{C}^{N \times M_r}$ is the phase of WTOs in $\mathbf{S}_g^R$. $\mathbf{x}_{g,m} = \tilde{\mathbf{F}}_K \tilde{\mathbf{s}}_{g,m}$, where $\tilde{\mathbf{s}}_{g,m} = \left[ \mathbf{s}_{g,m}^T \quad \mathbf{0}_{(N-1)\times 1}^T \right]^T$, and $\tilde{\mathbf{F}}_K$ is the $K(K = 2N - 1)$ point discrete Fourier transform (DFT) matrix.

### 3.1.2. JRC-CWG Design Algorithm

In this section, an algorithm based on the framework of the FR-CG algorithm and the FFT algorithm is applied to solve (10).

Define

$$
D(\boldsymbol{\Theta}_g) = \left[ \sum_{m=1}^{M} \mathbf{x}_{g,m} \odot \mathbf{x}_{g,m}^* \right]^H \left[ \sum_{m=1}^{M} \mathbf{x}_{g,m} \odot \mathbf{x}_{g,m}^* \right]
\tag{11}
$$

To solve (10), the gradient of $D(\boldsymbol{\Theta}_g)$ with respect to $\boldsymbol{\Theta}_g$ is required to be derived. To achieve this, the gradient of $D(\boldsymbol{\Theta}_g)$ with respect to $\theta_{g,m}(n)$ $(m \leq M_r)$ is derived first. According to the definition of the gradient of the real-valued function with respect to complex variables in [44], the gradient of $D(\boldsymbol{\Theta}_g)$ with respect to $\theta_{g,m}(n)$ is represented as

$$
\begin{aligned}
\frac{\partial D(\boldsymbol{\Theta}_g)}{\partial \theta_{g,m}(n)} &= \frac{\partial D(\boldsymbol{\Theta}_g)}{\partial s_{g,m}(n)} \cdot \frac{\partial s_{g,m}(n)}{\partial \theta_{g,m}(n)} + \frac{\partial D(\boldsymbol{\Theta}_g)}{\partial s_{g,m}^*(n)} \cdot \frac{\partial s_{g,m}^*(n)}{\partial \theta_{g,m}(n)} \\
&= j \frac{\partial D(\boldsymbol{\Theta}_g)}{\partial s_{g,m}(n)} \cdot s_{g,m}(n) - j \frac{\partial D(\boldsymbol{\Theta}_g)}{\partial s_{g,m}^*(n)} \cdot s_{g,m}^*(n)
\end{aligned}
\tag{12}
$$

In (12), $j$ represents the imaginary unit. According to the chain rule, we can determine that

$$
\begin{aligned}
\frac{\partial D(\boldsymbol{\Theta}_g)}{\partial s_{g,m}(n)} &= \sum_{k=1}^{K} \frac{\partial D(\boldsymbol{\Theta}_g)}{\partial \gamma_g(k)} \frac{\partial \gamma_g(k)}{\partial x_{g,m}(k)} \frac{\partial x_{g,m}(k)}{\partial s_{g,m}(n)} \\
&= \sum_{k=1}^{K} \gamma_g^*(k) x_{g,m}^*(k) \tilde{\mathbf{F}}_K(k, n)
\end{aligned}
\tag{13}
$$

and

$$
\begin{aligned}
\frac{\partial D(\boldsymbol{\Theta}_g)}{\partial s_{g,m}^*(n)} &= \sum_{k=1}^{K} \frac{\partial D(\boldsymbol{\Theta}_g)}{\partial \gamma_g^*(k)} \frac{\partial \gamma_g^*(k)}{\partial x_{g,m}^*(k)} \frac{\partial x_{g,m}^*(k)}{\partial s_{g,m}^*(n)} \\
&= \sum_{k=1}^{K} \gamma_g(k) x_{g,m}(k) \tilde{\mathbf{F}}_K^*(k, n)
\end{aligned}
\tag{14}
$$

In (13) and (14), $\gamma_g(k)$ is the $k$-th element of $\boldsymbol{\gamma}_g$, and $\boldsymbol{\gamma}_g = \sum_{m=1}^{M} \mathbf{x}_{g,m} \odot \mathbf{x}_{g,m}^*$. From (13) and (14), we can determine that

$$
\frac{\partial D(\boldsymbol{\Theta}_g)}{\partial s_{g,m}(n)} = \left[ \frac{\partial D(\boldsymbol{\Theta}_g)}{\partial s_{g,m}^*(n)} \right]^*
\tag{15}
$$

From (12) to (15), the gradient of $D(\boldsymbol{\Theta}_g)$ with respect to $\theta_{g,m}$ can be represented as

$$
\begin{aligned}
\frac{\partial D(\boldsymbol{\Theta}_g)}{\partial \theta_{g,m}} &= \left\{ \mathbf{V}\left[ \tilde{\mathbf{F}}_K \left( 2\boldsymbol{\gamma}_g \odot \mathbf{x}_{g,m}^* \right) \right] \right\} \odot \left( j\mathbf{s}_{g,m} \right) \\
&+ \left\{ \mathbf{V}\left[ \tilde{\mathbf{F}}_K^H \left( 2\boldsymbol{\gamma}_g \odot \mathbf{x}_{g,m} \right) \right] \right\} \odot \left( -j\mathbf{s}_{g,m}^* \right) \\
&= 2\mathrm{Re}\left\{ \left\{ \mathbf{V}\left[ \tilde{\mathbf{F}}_K \left( 2\boldsymbol{\gamma}_g \odot \mathbf{x}_{g,m}^* \right) \right] \right\} \odot \left( j\mathbf{s}_{g,m} \right) \right\}
\end{aligned}
\tag{16}
$$

where $\mathbf{V} = \left[ \mathbf{I}_N, \mathbf{0}_{N \times (N-1)} \right]$. Using (16), the gradient of $D(\boldsymbol{\Theta}_g)$ with respect to $\boldsymbol{\Theta}_g$ can be calculated, and the FR-CG algorithm can be applied to solve (10). Note that (10) and (16) can be calculated with the FFT algorithm to reduce computational complicity. The detailed algorithm for designing the $g$-th JRC-CWG is summarized in Algorithm 1.

---

**Algorithm 1:** The JRC-CWG design algorithm.

---

**Input:** $G$, $M_c$, $M_r$, $N$, the threshold $\varepsilon$, and WCIs.

**Step 1:** Initialize $\boldsymbol{\Theta}_g^{(0)} = \left[ \theta_{g,1}^{(0)}, \ldots, \theta_{g,M_r}^{(0)} \right]$, let $k_0 = 0$, $\alpha_0 = 0$, and the search direction $\tilde{\mathbf{V}}_d^{(0)} = \mathbf{0}_{N \times M_r}$.

**Step 2:** Calculate the gradient matrix $\tilde{\mathbf{G}}_g^{(k_0)} = \left[ \tilde{\mathbf{g}}_{g,1}^{(k_0)}, \tilde{\mathbf{g}}_{g,2}^{(k_0)}, \ldots \tilde{\mathbf{g}}_{g,M_r}^{(k_0)} \right]$ using (16), where $\tilde{\mathbf{g}}_{g,m}^{(k_0)} = \partial D\left[ \boldsymbol{\Theta}_g^{(k_0)} \right] / \partial \theta_{g,m}^{(k_0)}$.

**Step 3:** Calculate the search direction $\tilde{\mathbf{V}}_d^{(k_0+1)} = -\tilde{\mathbf{G}}_g^{(k_0)} + \alpha_{k_0} \tilde{\mathbf{V}}_d^{(k_0)}$.

**Step 4:** Compute $\boldsymbol{\Theta}_g^{(k_0+1)} = \boldsymbol{\Theta}_g^{(k_0)} + \lambda_{k_0} \tilde{\mathbf{V}}_d^{(k_0+1)}$, where $\lambda_{k_0}$ is the optimal solution that minimizes $D\left[ \boldsymbol{\Theta}_g^{(k_0+1)} \right]$, which is obtained with the line search method [45].

**Step 5:** If $\left| D\left[ \boldsymbol{\Theta}_g^{(k_0+1)} \right] - D\left[ \boldsymbol{\Theta}_g^{(k_0)} \right] \right| \leq \varepsilon$, go to **Output**,

　**else** (1) Let $k_0 = k_0 + 1$;

(2) Calculate the gradient matrix $\tilde{\mathbf{G}}_g^{(k_0)} = \left[ \tilde{\mathbf{g}}_{g,1}^{(k_0)}, \tilde{\mathbf{g}}_{g,2}^{(k_0)}, \ldots \tilde{\mathbf{g}}_{g,M_r}^{(k_0)} \right]$ using (16),

where $\tilde{\mathbf{g}}_{g,m}^{(k_0)} = \partial D\left[ \boldsymbol{\Theta}_g^{(k_0)} \right] / \partial \theta_{g,m}^{(k_0)}$;

(3) Calculate $\alpha_{k_0} = \left\| \tilde{\mathbf{G}}_g^{(k_0)} \right\|_F^2 / \left\| \tilde{\mathbf{G}}_g^{(k_0-1)} \right\|_F^2$;

(4) Turn to **Step 3**.

**Output:** The phase of WTOs $\boldsymbol{\Theta}_g^{(k_0+1)}$.

---

### 3.1.3. Computational Complexity Analysis

For the proposed algorithm to design the JRC-CWG in Algorithm 1, the computational complexity of step 2 is $O(M_r(M+1)K \log_2(K))$. The computational cost of step 3 is $O(NM_r)$. In step 4, the computational complexity is $O(\tilde{N}MK \log_2(K))$, where $\tilde{N}$ is the search time to obtain $\lambda_{k_0}$ with the line search method. In step 5 (2), the computational complexity is $O(M_r(M+1)K \log_2(K))$, and the computational cost of step 5 (3) is $O(NM_r)$. The total computational complexity of each iteration of the JRC-CWG design algorithm is approximately $O\left( M_r(M+1)K \log_2(K) + \tilde{N}MK \log_2(K) + 2NM_r \right)$. Hence, the computational complexity of the JRC-CWG design algorithm in Algorithm 1 is approximately $O\left( \tilde{L}M_r(M+1)K \log_2(K) + \tilde{L}\tilde{N}MK \log_2(K) + 2\tilde{L}NM_r \right)$, where $\tilde{L}$ is the number of iterations.

### 3.2. DR-JRC-CWG Design

In Section 3, we have designed the JRC-CWG under the assumption that the relative radial velocity $v$ between the JRC system and the target is known or can be ignored. However, the accurate $v$ is hard to be obtained, or $v$ cannot be ignored in practice. The range sidelobe level of the received JRC-CWG will increase if $v$ cannot be compensated at the radar receiver. To decrease the range sidelobe level of the received JRC-CWG with Doppler shift, a Doppler-resilient JRC-CWG, i.e., the DR-JRC-CWG is devised in this section.

3.2.1. Doppler Sensitivity Analysis

According to (3), the sum of matched filter outputs of the $g$-th JRC-CWG with Doppler shift $f_d$ is

$$y_g^r(n, f_d) = \delta_g \rho \sum_{m=1}^{M} e^{j2\pi(m-1)f_d T} \mathbf{s}_{g,m}^H \mathbf{U}_n \mathbf{s}_{g,m} + \sum_{m=1}^{M} \mathbf{n}'_{g,m} \tag{17}$$

As shown in (17), the Doppler frequency $f_d$ will affect the range sidelobe level of the JRC-CWG. It is necessary to design the DR-JRC-CWG when $f_d$ cannot be accurately acquired.

According to (17), the AF of the $g$-th JRC-CWG is defined as

$$y_g(n, \varphi_d) = \sum_{m=1}^{M} e^{j2\pi(m-1)\varphi_d} \mathbf{s}_{g,m}^H \mathbf{U}_n \mathbf{s}_{g,m}, \\ \text{for } n = 0, 1, \ldots, N-1 \tag{18}$$

and $y_g(n, \varphi_d) = y_g^*(-n, \varphi_d)$, for $n = 1-N, 2-N, \ldots, -1$, where $\varphi_d = f_d T$ is the normalized Doppler frequency.

Next, the AF of the $g$-th JRC-CWG is analyzed using the Taylor formula, and we will show how the nonzero-order Taylor expansion terms affect the Doppler resilience of the JRC-CWG. The Taylor expansion of the AF of the $g$-th JRC-CWG around $\varphi_d = 0$ can be represented as

$$y_g(n, \varphi_d) = \sum_{q=0}^{\infty} \frac{\sum_{m=1}^{M} (m-1)^q \mathbf{s}_{g,m}^H \mathbf{U}_n \mathbf{s}_{g,m}}{q!} (j2\pi\varphi_d)^q \tag{19}$$

Introduce $\boldsymbol{\beta}_{g,q} = \left[ \beta_{g,q}(1-N), \ldots, \beta_{g,q}(0), \ldots \beta_{g,q}(N-1) \right]^T$, where

$$\beta_{g,q}(n) = \frac{1}{q!} \sum_{m=1}^{M} (m-1)^q \mathbf{s}_{g,m}^H \mathbf{U}_n \mathbf{s}_{g,m} \\ \text{for } n = 0, 1, \ldots, N-1 \tag{20}$$

and $\beta_{g,q}(n) = \beta_{g,q}^*(-n)$, for $n = 1-N, 2-N, \ldots, -1$, and then, (19) can be rewritten as

$$y_g(n, \varphi_d) = \sum_{q=0}^{\infty} \beta_{g,q}(n)(j2\pi\varphi_d)^q \tag{21}$$

Let

$$P_{g,q}(n, \varphi_d) = \beta_{g,q}(n)(j2\pi\varphi_d)^q \tag{22}$$

Then, (21) can be rewritten as

$$y_g(n, \varphi_d) = \sum_{q=0}^{\infty} P_{g,q}(n, \varphi_d). \tag{23}$$

In (23), $y_g(n, \varphi_d)$ can be regarded as the summation of $P_{g,q}(n, \varphi_d)$ for different $q$. When $q = 0$, we will obtain

$$P_{g,0}(n, \varphi_d) = \sum_{m=1}^{M} \mathbf{s}_{g,m}^H \mathbf{U}_n \mathbf{s}_{g,m} \tag{24}$$

In (24), it can be seen that $P_{g,0}(n, \varphi_d)$ is the SACF of the $g$-th JRC-CWG. This indicates that only the zero-order Taylor expansion term is considered in the optimization problem (10). However, according to (23), if only the zero-order Taylor expansion term $P_{g,0}(n, \varphi_d)$ is considered in the JRC-CWG design, the nonzero-order Taylor expansion terms of $y_g$ will deteriorate the Doppler resilience of the JRC-CWGs. Hence, to suppress the sidelobe level of the JRC-CWG around $\varphi_d = 0$, more Taylor expansion terms of $y_g$, i.e., $P_{g,q}$, should be

optimized. Moreover, according to (20) and (22), as $q$ increases, the value of $P_{g,q}(n, \varphi_d)$ decreases because of the term $1/q!$. This means that some lower order of $P_{g,q}$ has the major effect on the Doppler resilience of the JRC-CWG. Furthermore, according to (22), when optimizing $P_{g,q}$, the term $(j2\pi\varphi_d)^q$ can be regarded as a constant. Hence, it is reasonable to optimize $P_{g,q}$ by optimizing $\boldsymbol{\beta}_{g,q}$. In summary, to design the DR-JRC-CWG, $\boldsymbol{\beta}_{g,q}$ for $q = 0, 1, \ldots, Q$, should be jointly optimized.

### 3.2.2. Problem Formulation

From the discussion above, the optimization problem to design the $g$-th DR-JRC-CWG can be represented as

$$
\begin{aligned}
\min_{\mathbf{S}_g^R} \quad & \sum_{q=0}^{Q} \tilde{w}_q \left( \frac{1}{\tilde{E}(q)} \boldsymbol{\beta}_{g,q}^H \boldsymbol{\beta}_{g,q} - 1 \right) \\
s.t. \quad & |s_{g,m}(n)| = \frac{1}{\sqrt{N}} \quad m = 1, 2, \ldots, M, \quad n = 1, 2, \ldots, N
\end{aligned}
\tag{25}
$$

where $\tilde{E}(q) = \frac{1}{q!} \cdot \sum_{m=1}^{M} (m-1)^q$, and $\tilde{w}_q$ is the weight satisfying $\sum_{q=0}^{Q} \tilde{w}_q = 1$. Similar to (10), to obtain $G$ optimal DR-JRC-CWGs, $G$ optimization problems similar to (25) should be solved in parallel. It can be seen that when $Q = 0$, the optimization problem in (25) will be the optimization problem in (9). Similar to (10), the optimization problem (25) can be transformed into

$$
\min_{\boldsymbol{\Theta}_g} \sum_{q=0}^{Q} \tilde{w}_q \left[ \sum_{m=1}^{M} (m-1)^q \mathbf{x}_{g,m} \odot \mathbf{x}_{g,m}^* \right]^H \left[ \sum_{m=1}^{M} (m-1)^q \mathbf{x}_{g,m} \odot \mathbf{x}_{g,m}^* \right]
\tag{26}
$$

### 3.2.3. DR-JRC-CWG Design Algorithm

For simplicity, let

$$
D_{dop}(\boldsymbol{\Theta}_g) = \sum_{q=0}^{Q} \tilde{w}_q h_{g,q}(\boldsymbol{\Theta}_g)
\tag{27}
$$

where $h_{g,q}(\boldsymbol{\Theta}_g) = \left[ \sum_{m=1}^{M} (m-1)^q \mathbf{x}_{g,m} \odot \mathbf{x}_{g,m}^* \right]^H \left[ \sum_{m=1}^{M} (m-1)^q \mathbf{x}_{g,m} \odot \mathbf{x}_{g,m}^* \right]$. The gradient of $D_{dop}(\boldsymbol{\Theta}_g)$ with respect to $\boldsymbol{\Theta}_g$ can be represented as

$$
\frac{\partial D_{dop}(\boldsymbol{\Theta}_g)}{\partial \boldsymbol{\Theta}_g} = \sum_{q=0}^{Q} \tilde{w}_q \frac{\partial h_{g,q}(\boldsymbol{\Theta}_g)}{\partial \boldsymbol{\Theta}_g}
\tag{28}
$$

The gradient of $h_{g,q}(\boldsymbol{\Theta}_g)$ with respect to $\boldsymbol{\theta}_{g,m}$ $(m \le M_r)$ is

$$
\frac{\partial h_{g,q}(\boldsymbol{\Theta}_g)}{\partial \boldsymbol{\theta}_{g,m}} = 2\mathrm{Re}\left\{ (m-1)^q \left\{ \mathbf{V} \left[ \tilde{\mathbf{F}}_K \left( 2\boldsymbol{\gamma}_{g,q} \odot \mathbf{x}_{g,m}^* \right) \right] \right\} \odot (j\mathbf{s}_{g,m}) \right\}
\tag{29}
$$

where $\boldsymbol{\gamma}_{g,q} = \sum_{m=1}^{M} \left[ (m-1)^q \mathbf{x}_{g,m} \odot \mathbf{x}_{g,m}^* \right]$. The derivation of (29) is similar to that in (16). In addition, (27) and (29) can be computed using the FFT algorithm to reduce the computational complexity. The DR-JRC-CWG design algorithm is summarized in Algorithm 2.

---

**Algorithm 2:** The DR-JRC-CWG design algorithm

---

**Input:** $G$, $M_c$, $M_r$, $N$, $\tilde{w}_q$, the threshold $\varepsilon$, $Q$, and WCIs.

**Step 1**: Initialize $\mathbf{\Theta}_g^{(0)} = \left[\mathbf{\theta}_{g,1}^{(0)}, \ldots, \mathbf{\theta}_{g,M_r}^{(0)}\right]$, and let $k_0 = 0$, $\alpha_0 = 0$, and $\tilde{\mathbf{V}}_d^{(0)} = \mathbf{0}_{N \times M_r}$.

**Step 2**: Calculate the gradient matrix $\tilde{\mathbf{G}}_g^{(k_0)} = \left[\tilde{\mathbf{g}}_{g,1}^{(k_0)}, \tilde{\mathbf{g}}_{g,2}^{(k_0)}, \ldots \tilde{\mathbf{g}}_{g,M_r}^{(k_0)}\right]$ using (28) and (29), where
$\tilde{\mathbf{g}}_{g,m}^{(k_0)} = \partial D\left[\mathbf{\Theta}_g^{(k_0)}\right] / \partial \mathbf{\theta}_{g,m}^{(k_0)}$.

**Step 3**: Calculate the search direction $\tilde{\mathbf{V}}_d^{(k_0+1)} = -\tilde{\mathbf{G}}_g^{(k_0)} + \alpha_{k_0} \tilde{\mathbf{V}}_d^{(k_0)}$.

**Step 4**: Compute $\mathbf{\Theta}_g^{(k_0+1)} = \mathbf{\Theta}_g^{(k_0)} + \lambda_{k_0} \tilde{\mathbf{V}}_d^{(k_0+1)}$, where $\lambda_{k_0}$ is the optimal solution that minimizes
$D_{dop}\left[\mathbf{\Theta}_g^{(k_0+1)}\right]$, which is obtained with the line search method.

**Step 5**: **If** $\left| D_{dop}\left[\mathbf{\Theta}_g^{(k_0+1)}\right] - D_{dop}\left[\mathbf{\Theta}_g^{(k_0)}\right] \right| \leq \varepsilon$, go to **Output**,
**else** (1) Let $k_0 = k_0 + 1$;
(2) Calculate the gradient matrix $\tilde{\mathbf{G}}_g^{(k_0)} = \left[\tilde{\mathbf{g}}_{g,1}^{(k_0)}, \tilde{\mathbf{g}}_{g,2}^{(k_0)}, \ldots \tilde{\mathbf{g}}_{g,M_r}^{(k_0)}\right]$ using (28) and
(29), where $\tilde{\mathbf{g}}_{g,m}^{(k_0)} = \partial D\left[\mathbf{\Theta}_g^{(k_0)}\right] / \partial \mathbf{\theta}_{g,m}^{(k_0)}$;
(3) Calculate $\alpha_{k_0} = \|\tilde{\mathbf{G}}_g^{(k_0)}\|_F^2 / \|\tilde{\mathbf{G}}_g^{(k_0-1)}\|_F^2$;
(4) Turn to **Step 3**.

**Output:** The phase of WTOs $\mathbf{\Theta}_g^{(k_0+1)}$.

---

### 3.2.4. Computational Complexity Analysis

For the DR-JRC-CWG design algorithm in Algorithm 2, the computational complexity of step 2 is $O\big((Q+1)M_r(M+1)K\log_2(K)\big)$. The computational cost of step 3 is $O(NM_r)$. The computational complexity of step 4 is $O\big((Q+1)\tilde{N}MK\log_2(K)\big)$, where $\tilde{N}$ is the search time to obtain $\lambda_{k_0}$ with the line search method. Moreover, in step 5 (2), the computational complexity is $O\big((Q+1)M_r(M+1)K\log_2(K)\big)$, and the computational cost of step 5 (3) is $O(NM_r)$. Then, the total computational complexity of each iteration of the DR-JRC-CWG design algorithm is approximately $O\big((Q+1)\big(M_r(M+1)+\tilde{N}M\big)K\log_2(K)+2NM_r\big)$. Hence, the computational complexity of the DR-JRC-CWG design algorithm in Algorithm 2 is approximately $O\big(\tilde{L}(Q+1)\big(M_r(M+1)+\tilde{N}M\big)K\log_2(K)+2\tilde{L}NM_r\big)$, where $\tilde{L}$ is the number of iterations.

## 4. Results

In this section, several simulation results are given to evaluate the performances of the designed JRC-CWG and the DR-JRC-CWG. In the following, unless otherwise mentioned, the number of groups $G$ of the JRC-CWG or the DR-JRC-CWG is set to be 1. The peak-to-sidelobe ratio (PSLR) of SACF of the JRC-CWG and the DR-JRC-CWG is utilized to measure their sidelobe performance. The BPSK and the 16-PSK waveforms are employed as the WCIs for the designed JRC-CWG and DR-JRC-CWG. For the WCIs with BPSK modulation, the JRC-CWG is denoted as the JRC-CWG-BPSK, and the DR-JRC-CWG is denoted as DR-JRC-CWG-BPSK in the following. For the WCIs with 16-PSK modulation, the JRC-CWG and the DR-JRC-CWG are denoted as the JRC-CWG-16-PSK and the DR-JRC-CWG-16-PSK, respectively.

### 4.1. Performance of the JRC-CWG

Let the number of WCIs $M_c$ in each JRC-CWG be fixed to 1, and the sidelobe performance of the JRC-CWG with different numbers of waveforms $M$ used in each JRC-CWG is shown in Figure 5. The sidelobe performances of the two other JRC waveforms, i.e., the LFM-BPSK waveform in [31] and the OPP waveform in [13], are also shown in Figure 5. In Figure 5, the OPP-BPSK denotes the OPP waveforms modulated by the BPSK waveforms. The length of each JRC waveform in Figure 5 is $N = 64$. To measure the sidelobe performance of the LFM-BPSK waveforms and the OPP-BPSK waveforms, the PSLR of SACF of $M$ different LFM-BPSK waveforms and $M$ different OPP-BPSK waveforms are calculated, respectively.

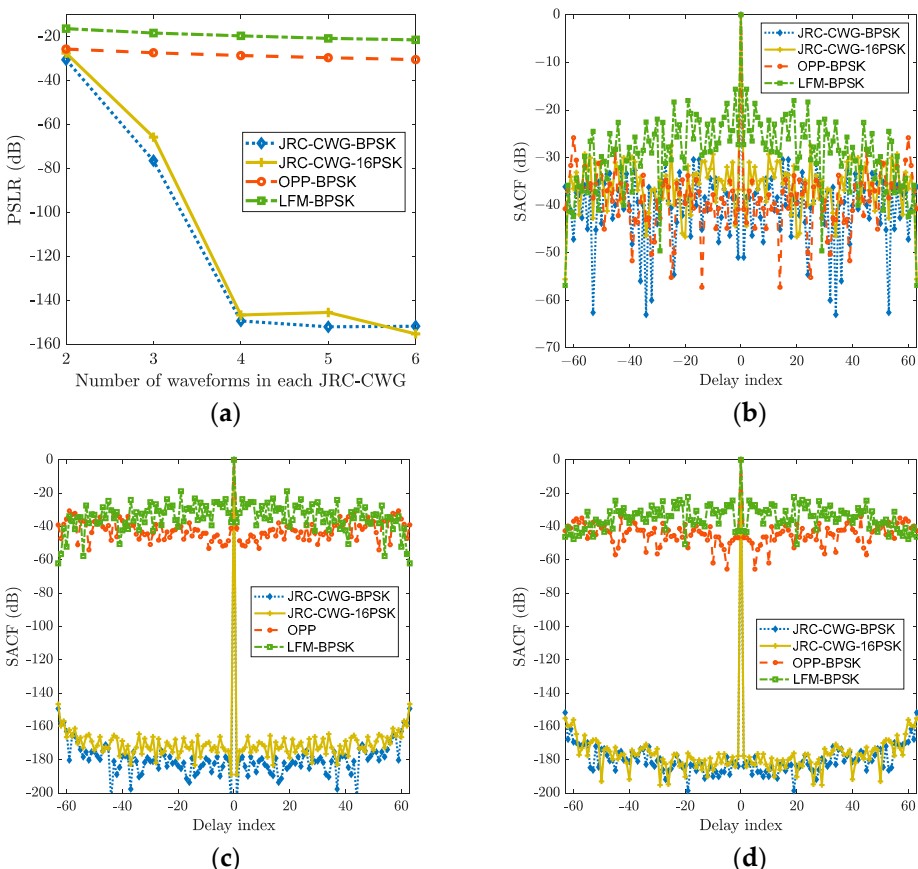

**Figure 5.** (**a**) PSLRs of the JRC waveforms with different number of waveforms *M* in each JRC-CWG. (**b**) SACF with *M* = 2. (**c**) SACF with *M* = 4. (**d**) SACF with *M* = 6.

In Figure 5a, the PSLRs of the JRC waveforms versus *M* are given. Moreover, the SACFs of the JRC waveforms with the number of waveforms *M* =2,4,6 are shown, respectively, in Figure 5b–d. In Figure 5a, when *M* increases, the PSLR of all JRC waveforms decreases, and the PSLRs of the designed JRC-CWG-BPSK and the JRC-CWG-16-PSK decrease faster than those of the other two JRC waveforms. Specifically, when *M* = 2, the PSLR of the JRC-CWG-BPSK is 4.8 dB lower than that of the OPP-BPSK waveform, and 14.1 dB lower than that of the LFM-BPSK waveform. When *M* = 2, the PSLR of the designed JRC-CWG-16-PSK is 3.9 dB lower than that of the OPP-BPSK waveform and 13.2 dB lower than that of the LFM-BPSK waveform. When *M* = 4, the PSLRs of the designed JRC-CWG-BPSK and JRC-CWG-16-PSK are at least 120 dB lower than those of the OPP-BPSK waveform and the LFM-BPSK waveform. As depicted in Section 2, $M = M_r + M_c$, where $M_r$ is the number of WTOs employed in each JRC-CWG. Thus, $M_r$ will increase with the increasing *M* when $M_c$ is fixed. With the increase in $M_r$, more degrees of freedom will be employed to suppress the sidelobe of the JRC-CWG SACF according to (9), which will lead to a lower PSLR than other JRC waveforms. Due to the limitation of the computation accuracy of the used computer, the PSLR of the designed JRC-CWG is approximately −150 dB and will not decrease anymore when $M \geq 4$. Furthermore, it can be seen in Figure 5 that the sidelobe performance of the designed JRC-CWG-BPSK is close to that of the designed JRC-CWG-16-PSK with different *M*. That is to say, the modulation approach has little influence on the sidelobe performance of the designed JRC-CWG.

From Figure 5, we can see that with a sufficiently large *M*, the sidelobe level of the designed JRC-CWG can be sufficiently suppressed. However, in practice, a large *M* is usually not recommended. With a large *M*, it will take a longer time to transmit one JRC-CWG, and the coefficient $\delta_g$ in (2) may be changed. In practice, the *M* should make sure that the coefficient $\delta_g$ does not change during the CPI.

### 4.2. Performance of the DR-JRC-CWG

In this subsection, the performance of DR-JRC-CWG will be shown. The AF defined in (18) and Delay-Doppler maps of the designed JRC-CWG-BPSK are shown in Figure 6(a1,a2). Let $\mathbf{w} = \left[\tilde{w}_0, \tilde{w}_1, \ldots, \tilde{w}_Q\right]^T$ collect the weights in (26), and the AFs and Delay-Doppler maps of the designed DR-JRC-CWG-BPSK and DR-JRC-CWG-16-PSK with different weights $\mathbf{w}$ are shown in Figure 6(b1–d2). Furthermore, the AFs and Delay-Doppler maps of the LFM-BPSK waveform and OPP-BPSK waveform are shown in Figure 6(e1–f2). The simulation parameters used in Figure 6 are listed in Table 1.

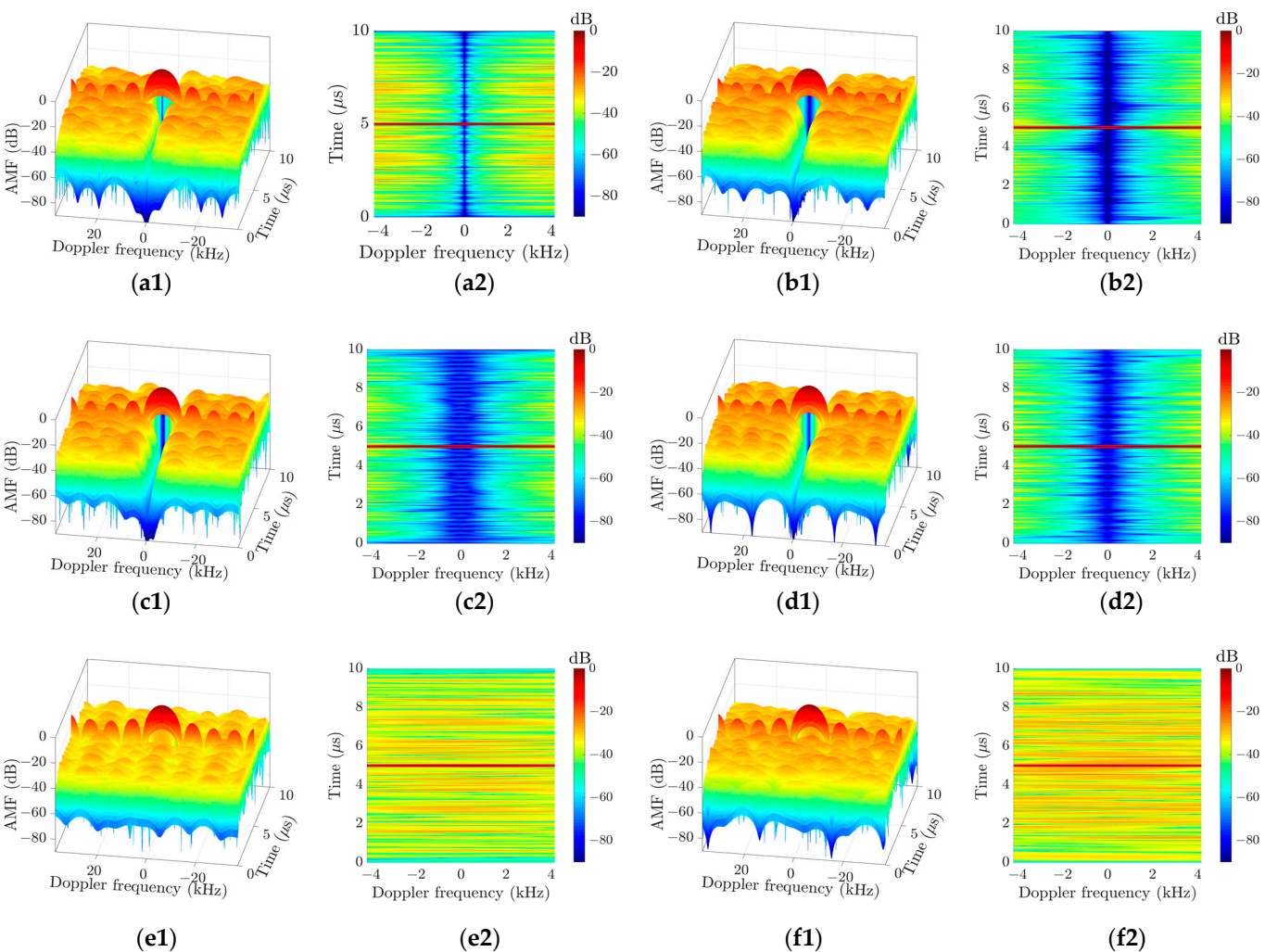

**Figure 6.** AFs and Delay-Doppler maps of the JRC waveforms. (**a1**) AF of the JRC-CWG-BPSK; (**a2**) Delay-Doppler map of the JRC-CWG-BPSK; (**b1**) AF of the DR-JRC-CWG-BPSK with $\mathbf{w} = \mathbf{w}_1$; (**b2**) Delay-Doppler map of the DR-JRC-CWG-BPSK with $\mathbf{w} = \mathbf{w}_1$; (**c1**) AF of the DR-JRC-CWG-BPSK with $\mathbf{w} = \mathbf{w}_2$; (**c2**) Delay-Doppler map of the DR-JRC-CWG-BPSK with $\mathbf{w} = \mathbf{w}_2$; (**d1**) AF of the DR-JRC-CWG-16-PSK with $\mathbf{w} = \mathbf{w}_2$; (**d2**) Delay-Doppler map of the DR-JRC-CWG-16-PSK with $\mathbf{w} = \mathbf{w}_2$; (**e1**) AF of the OPP-BPSK waveform; (**e2**) Delay-Doppler map of the OPP-BPSK waveform; (**f1**) AF of the LFM-BPSK waveform; and (**f2**) Delay-Doppler map of the LFM-BPSK waveform.

**Table 1.** Parameters used in Figure 6.

| Figure | JRC Waveform | $Q$ | $w$ | $M_c$ | $M$ | $T_p$ | PRI | $N$ |
|---|---|---|---|---|---|---|---|---|
| Figure 6(a1,a2) | JRC-CWG-BPSK | 0 | - | 1 | 6 | 5 µs | 20 µs | 64 |
| Figure 6(b1,b2) | DR-JRC-CWG-BPSK | 1 | $\mathbf{w}_1 = [0.95, 0.05]^T$ | 1 | 6 | 5 µs | 20 µs | 64 |
| Figure 6(c1,c2) | DR-JRC-CWG-BPSK | 1 | $\mathbf{w}_2 = [0.9, 0.1]^T$ | 1 | 6 | 5 µs | 20 µs | 64 |
| Figure 6(e1,e2) | DR-JRC-CWG-16-PSK | 1 | $\mathbf{w}_2 = [0.9, 0.1]^T$ | 1 | 6 | 5 µs | 20 µs | 64 |
| Figure 6(e1,e2) | OPP-BPSK | - | - | - | 6 | 5 µs | 20 µs | 64 |
| Figure 6(f1,f2) | LFM-BPSK | - | - | - | 6 | 5 µs | 20 µs | 64 |

From Figure 6, we can see that all the AFs have a 'thumbtack' shape. It can be seen that there is a valley in the AFs of the designed JRC-CWG-BPSK, DR-JRC-CWG-BPSK, and DR-JRC-CWG-16-PSK around $f_d = 0$, while there is no valley in the AFs of the LFM-BPSK waveform and the OPP-BPSK waveform. This means that the designed JRC-CWG-BPSK, DR-JRC-CWG-BPSK, and DR-JRC-CWG-16-PSK have better PSLR performances around $f_d = 0$ than the other two JRC waveforms. Moreover, the valley indicates that the PSLRs of the JRC-CWG-BPSK, DR-JRC-CWG-BPSK, and DR-JRC-CWG-16-PSK increase as $f_d$ increases, which is consistent with the analysis of (17). Compared with Figure 6(a2), the valley in Figure 6(b2,c2,d2) is much wider. The wider the valley is, the more slowly the PSLR will increase with the increase in $f_d$. Therefore, compared with the designed JRC-CWG-BPSK, the devised DR-JRC-CWG-BPSK and DR-JRC-CWG-16-PSK have better Doppler resilience. Furthermore, compared with Figure 6(b2), the valley in Figure 6(c2) is wider, which indicates that the Doppler resilience of the designed DR-JRC-CWG-BPSK will be improved when the weight $\tilde{w}_1$ in $\mathbf{w}$ is increased. From Figure 6, we can see that compared with the LFM-BPSK waveform, OPP-BPSK waveform, and the JRC-CWG, the proposed DR-JRC-CWG has the best Doppler resilience. This indicates that the designed DR-JRC-CWG is more suitable to be employed by automotive JRC systems than other JRC waveforms when $f_d$ cannot be obtained or ignored.

In Figure 7, the relationship between the PSLR and Doppler frequency is shown. The simulation parameters in Figure 7 are the same as those in Figure 6. In accordance with the IEEE 802.11p standard, we assume that the carrier frequency of the JRC waveforms in Figure 7 is 5.9 GHz [46], i.e., the wavelength $\lambda$ is 0.051 m. In Figure 7, the horizontal line for $-35$ dB is plotted. To make sure the PSLRs of the JRC waveforms are lower than $-35$ dB, the maximum Doppler frequencies and the corresponding radial velocities of the target in Figure 7 are listed in Table 2.

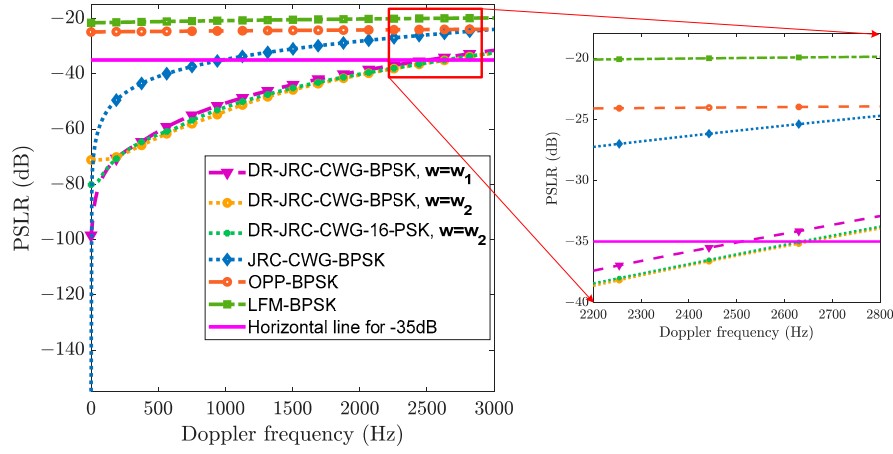

**Figure 7.** Relation curves between PSLR and Doppler frequency.

**Table 2.** PSLRs of the JRC waveforms and the corresponding velocities.

| JRC Waveform | $w$ | PSLR (dB) | $f_d$ (Hz) | Velocity (m/s) |
|---|---|---|---|---|
| JRC-CWG-BPSK | - | $-35$ | 986.3 | 25.1 |
| DR-JRC-CWG-BPSK | $\mathbf{w}_1 = [0.95, 0.05]^T$ | $-35$ | 2513.0 | 63.9 |
| DR-JRC-CWG-BPSK | $\mathbf{w}_2 = [0.9, 0.1]^T$ | $-35$ | 2654.0 | 67.5 |
| DR-JRC-CWG-16-PSK | $\mathbf{w}_2 = [0.9, 0.1]^T$ | $-35$ | 2631.4 | 66.9 |
| OPP-BPSK | - | $-35$ | NA | NA |
| LFM-BPSK | - | $-35$ | NA | NA |

In Table 2, the NA stands for not-applicable. It can be determined that if the PSLR of the JRC waveform is required to be lower than $-35$ dB, the OPP-BPSK waveform and the LFM-BPSK waveform will not be applicable. To make sure the PSLR is lower than $-35$ dB, the radial velocity $v$ between the JRC platform and the target should satisfy that $v < 25.1$ m/s for JRC-CWG-BPSK, $v < 63.9$ m/s for DR-JRC-CWG-BPSK with $\mathbf{w} = \mathbf{w}_1$, $v < 67.5$ m/s for the DR-JRC-CWG-BPSK with $\mathbf{w} = \mathbf{w}_2$, and $v < 66.9$ m/s for the DR-JRC-CWG-16-PSK with $\mathbf{w} = \mathbf{w}_2$, respectively. Hence, the designed DR-JRC-CWGs have better Doppler resilience than the designed JRC-CWGs. Compared with the DR-JRC-CWG-BPSK with $\mathbf{w} = \mathbf{w}_1$, i.e., $\tilde{w}_1 = 0.05$, and $\mathbf{w} = \mathbf{w}_2$, i.e., $\tilde{w}_1 = 0.1$, it can be observed that the Doppler resilience of the designed DR-JRC-CWG-BPSK will be improved when the weight $\tilde{w}_1$ in $\mathbf{w}$ is increased. Moreover, the designed DR-JRC-CWG-BPSK with $\mathbf{w} = \mathbf{w}_2$ has a similar performance to the designed DR-JRC-CWG-16-PSK with $\mathbf{w} = \mathbf{w}_2$. From Figure 7, we can see that the designed DR-JRC-CWG has better Doppler resilience than the OPP-BPSK waveform, the LFM-BPSK waveform, and the JRC-CWG. Moreover, the Doppler resilience of the designed DR-JRC-CWG satisfies the demand of automotive JRC systems, while that of other JRC waveforms does not.

In Figure 7, compared with the designed JRC-CWG-BPSK, the PSLR of the designed DR-JRC-CWG-BPSK at $f_d = 0$ is larger. Moreover, the larger the weight $\tilde{w}_1$ in $\mathbf{w}$ is, the larger the PSLR of the designed DR-JRC-CWG-BPSK at $f_d = 0$ is. Specifically, the PSLR of the designed JRC-CWG-BPSK at $f_d = 0$ is $-151.9$ dB. For the designed DR-JRC-CWG-BPSK with $\mathbf{w} = \mathbf{w}_1$, and the DR-JRC-CWG-BPSK with $\mathbf{w} = \mathbf{w}_2$, the PSLR at $f_d = 0$ is $-98.5$ dB and $-71.2$ dB, respectively. Note that in the multi-objective optimization problem (25), the Doppler resilience and the sidelobe level of SACF of the DR-JRC-CWG are jointly optimized, whereas in the optimization problem (9) only the sidelobe level of the SACF of the JRC-CWG is optimized. Hence, the PSLR of the DR-JRC-CWG may be higher than that of the JRC-CWG. Moreover, the larger the weight $\tilde{w}_1$ in $\mathbf{w}$ is, the higher the PSLR will be.

From Figure 7, it can be seen that the DR-JRC-CWG has a worse sidelobe performance than the JRC-CWG at $f_d = 0$. In practice, the user can employ the DR-JRC-CWG at first to estimate the velocity of the target. Then, the JRC-CWG can be employed, since the Doppler shift can be compensated at the radar receiver by using the estimated velocity.

Generally, an ITS often suffers serious clutter. In this scenario, the estimated target output signal-to-noise ratios (SNRs) of the JRC waveforms with different Doppler frequencies are illustrated in Figure 8. The simulation parameters of the JRC waveforms in Figure 8 are the same as those in Figure 7. The SNR is estimated by using the SNR estimation method in [47]. The input SNR of the JRC system is 0 dB in Figure 8. It can be seen that as the Doppler frequency increases, the output SNR of all the JRC waveforms decreases. However, the output SNR of the designed DR-JRC-CWG decreases slower than that of the other JRC waveforms. Furthermore, the output SNRs of the designed JRC-CWG and the DR-JRC-CWG are higher than those of the OPP-BPSK and the LFM-BPSK waveforms when $f_d = 0$. Hence, the DR-JRC-CWG has the best performance in terms of the output SNR.

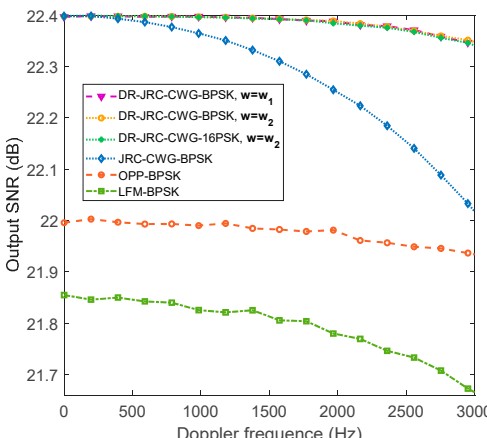

**Figure 8.** Output SNRs of the JRC waveforms at the radar receiver with different Doppler frequencies.

*4.3. Communications Performance*

In this subsection, the BER and communications rate are applied to evaluate the communications performance of the JRC waveforms.

The variations in the BER with the bit energy-to-noise power spectral density ratios (which is denoted as $E_b/N_0$ in Figure 9) of the JRC waveforms with the BPSK and 16-PSK modulation schemes are shown in Figure 9a,b, respectively. Moreover, the BERs of the traditional BPSK waveform and the traditional 16-PSK waveform are also illustrated in Figure 9. The OPP-16-PSK in Figure 9b denotes the OPP waveforms with 16-PSK modulation. In Figure 9a, we can see that both the designed JRC-CWG-BPSK and DR-JRC-CWG-BPSK have the same BER as the traditional BPSK waveforms. To make sure the BER is lower than $10^{-5}$, the $E_b/N_0$ should be greater than 9.6 dB for the JRC-CWG-BPSK and DR-JRC-CWG-BPSK, greater than 11.1 dB for the OPP-BPSK waveform, and greater than 12.4 dB for the LFM-BPSK waveform. For the JRC waveforms with 16-PSK modulation in Figure 9b, the designed JRC-CWG-16-PSK and DR-JRC-CWG-16-PSK have the same BER performance as the traditional 16-PSK waveforms. To ensure that the BER is lower than $10^{-5}$, the $E_b/N_0$ should be greater than 17.5 dB for the designed JRC-CWG-16PSK and DR-JRC-CWG-16PSK, and greater than 23.4 dB for the OPP-16-PSK. Obviously, compared with the other two JRC waveforms, the designed JRC-CWG and DR-JRC-CWG have the best communications performances in terms of BER for the same modulation scheme.

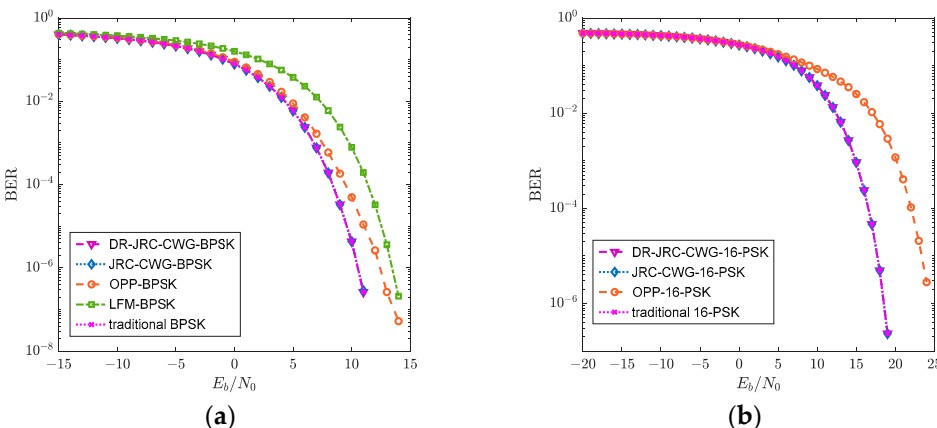

**Figure 9.** (**a**) BERs of the JRC waveforms with BPSK modulation; (**b**) BERs of the JRC waveforms with 16-PSK modulation.

The communications rates of the JRC-CWG and the DR-JRC-CWG are given in (7). It can be seen from (7) that the communications rate of the JRC-CWG or the DR-JRC-CWG will be increased when the number of WCIs in each JRC-CWG or DR-JRC-CWG $M_c$ is

increased. Let $M$ be a fixed value of 16, the PRI $T$ be 25 $\mu$s, and the length of each waveform $N$ be 64; the relationship between the communications rate and the PSLR of the designed JRC-CWG-16-PSK and the DR-JRC-CWG-16-PSK is shown in Figure 10. According to (7), the communications rate of the designed JRC-CWG-16-PSK and DR-JRC-CWG-16-PSK is $R_b = 0.64M_c$ Mbps. It can be seen that for the designed JRC-CWG-16-PSK and DR-JRC-CWG-16-PSK, the PSLR increases as the communications rate or $M_c$ increases. Specifically, when the communications rate is 0.64 Mbps, i.e., $M_c = 1$, the PSLRs of the designed JRC-CWG-16-PSK and the DR-JRC-CWG-16-PSK are $-165.3$ dB and $-118.8$ dB, respectively. When the communications rate is increased to 9.6 Mbps, i.e., $M_c = 15$, the PSLRs of the designed JRC-CWG-16-PSK and the DR-JRC-CWG-16-PSK are increased to $-33.0$ dB and $-30.7$ dB, respectively. Since $M = M_r + M_c$, the number of WTOs $M_r$ in each JRC-CWG or DR-JRC-CWG will decrease when $M_c$ increases with $M$ being fixed. Since only the WTOs are optimized to improve the sidelobe performance of the designed JRC waveforms in the optimization problems (9) and (25), the degrees of freedom of the optimization problems will decrease when $M_r$ decreases, which will lead to a worse PSLR performance. It can be observed that there is a trade-off between the communications rate and the PSLR of the designed JRC-CWG-16-PSK and DR-JRC-CWG-16-PSK. In practice, when designing the JRC-CWG or the DR-JRC-CWG, the trade-off curve in Figure 10 can be used to satisfy the requirements of the PSLR and the communications rate.

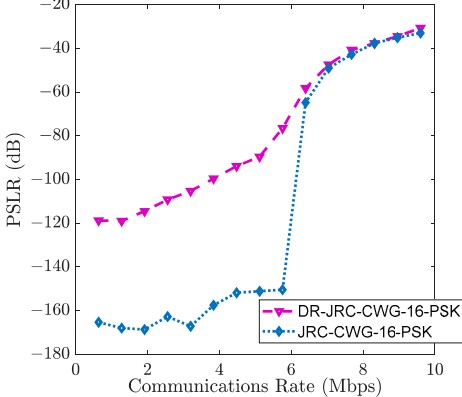

**Figure 10.** PSLRs of the designed JRC-CWG-16-PSK and DR-JRC-CWG-16-PSK versus communications rates.

The maximum communications rates and the corresponding PSLRs of the JRC waveforms are listed in Table 3. The simulation parameters used in Table 3 are the same as those in Figure 10. Although the OPP-16-PSK has a slightly higher communications rate than the designed JRC-CWG-16-PSK and DR-JRC-CWG-16-PSK, it has much worse performance in terms of PSLR and BER according to Figure 9b.

**Table 3.** Maximum communications rates and the corresponding PSLRs of the JRC waveforms.

| JRC Waveform | Communications Rate (Mbps) | PSLR (dB) |
| --- | --- | --- |
| DR-JRC-CWG-16-PSK | 9.6 | $-30.72$ |
| JRC-CWG-16-PSK | 9.6 | $-33.01$ |
| OPP-16-PSK | 10.24 | $-17.33$ |
| LFM-BPSK | 2.56 | $-19.27$ |

## 5. Discussion

In this paper, the JRC-CWG and the DR-JRC-CWG design methods were proposed, which have low sidelobe levels and good communications performances. The designed JRC-CWG and the DR-JRC-CWG can be used for automotive JRC systems. However, the proposed methods have to make a trade-off between the radar and communications perfor-

mance. In our future work, we will try to avoid this compromise. Moreover, polarization is an important characteristic that can be considered in waveform design to improve the performance of JRC waveforms [38]. The multiple-input multiple-output (MIMO) technique is not only used in radar but also in communications, and can also be employed in JRC-CWG design in our future work.

## 6. Conclusions

In this paper, we designed a JRC waveform group based on CSSs called the JRC-CWG. In order to achieve better radar performance, the JRC-CWG was optimized to obtain the SACF with a low sidelobe level. However, it has poor Doppler resilience. To improve the Doppler resilience of the JRC-CWG, the DR-JRC-CWG design method was proposed. To devise the JRC-CWG and DR-JRC-CWG, the algorithm based on the framework of the FR-CG algorithm and the FFT algorithm was employed, which can be implemented using FFT to decrease the calculation complexity. Compared with the OPP waveform and LFM-BPSK waveform, the designed JRC-CWG and DR-JRC-CWG have favorable radar and communications performances.

**Author Contributions:** Conceptualization, H.L., Y.L. and G.L.; writing—original draft preparation, H.L.; writing—review and editing, Y.L. and Y.C.; supervision, G.L.; funding acquisition, Y.L. and G.L.; All authors have read and agreed to the published version of the manuscript.

**Funding:** This work was supported by the National Natural Science Foundation of China (NSFC) under Grants 62001352 and 61931016, by the Fundamental Research Funds for the Central Universities under grant no. XJS220207, and by the Foundation for Innovative Research Groups of the National Natural Science Foundation of China under grant no. 61621005.

**Data Availability Statement:** The data presented in this study are available on request from the corresponding author.

**Conflicts of Interest:** The authors declare no conflict of interest.

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
