# Peer review of "Joint Radar and Communications Waveform Design Based on Complementary Sequence Sets"

_remotesensing, doi:10.3390/rs15030645_

Round 1

Reviewer 1 Report (New Reviewer)

Authors report on the waveform design and numerical results of the  automotive joint radar and communication. This paper is well written and  well organised. But, the following points should be addressed before the manuscript is considered for publication:

- In the case of radar receiver model, the factors to consider have been simplified too much, for example, the complex scattering coefficients as a constnat. When building up the radar model, the target modeling such as radar cross section, scattering characteristics(Swerling cases), and so on is very important. So, your approach is good, but the effectiveness of your signal model is not appropriate for the real world. Practical real world-based modeling is needed.

- The calculation time and cost using FFT algorithm including iteration time for optimization are dependent on the amount of the points for FFT. As the amount of FFT calculation increases, the system should be bulky because of the increase of signal processing hardware such as FPGAs. So, this would be not applicable to automotive area. Please give me your opinion about this point.

- Why didn't you consider or investigate the polarization of the waveform for radar and communication signal modeling?

Author Response

Reviewer 2 Report (New Reviewer)

A JRC waveform is designed based on CSS to improve the peak-to-sidelobe ratio for radar sensing. Overall, the paper is well written with comprehensive illustration and validation. 

A minor comment is about the trade-off between sensing performance and communication data rate. It seems that using CSS can reduce the communication data rate. This is worth more illustration in the work. I also suggest changing the x-axis of Fig. 9 to Eb/N0 to obtain a fair comparison of different schemes. 

Author Response

Reviewer 3 Report (New Reviewer)

In this paperthe authors claimed to propose two joint radar and communications (JRCwaveform design methods based on the philosophy of complementary sequence. It is an interesting idea to apply the complementary sequence set (CSS) into JRC waveform design to optimize the performance of radar and communications. The paper is well described, and there are some suggestions from the reviewer.

1. In terms of writing regularity, there is a comma missed at the end of (29)a period is lost at the end of the title of Table 4, and a comma is redundant in (20).

2. Please use the same font size for the tables. It can be seen that the Table 1~2 have the different font size from the Table 3~5.

3. I wonder if the word ‘Flecher-Reeves’ should be ‘Fletcher-Reeves’ instead.

4. In reference, there is a typo in reference 1 that the dash is discrete. I suggest the author to review the reference once again.

5. The English writing should be improved. For instance, in Section 5, ‘For the WCIs with BPSK modulation…. For the WCIs with 16-PSK modulation, …’. These two sentences need to be polished since they seem to be much too similar. 

6. In Introduction, the author should show the contribution of the work compared with the state-of-the-art.

7. In (20) of the paper,  is defined. However, is also used in Table 1 and 2 as a step length. Please change it to avoid symbol misleading.

Author Response

This manuscript is a resubmission of an earlier submission. The following is a list of the peer review reports and author responses from that submission.

Round 1

Reviewer 1 Report

This paper focuses on joint radar and communications waveform design. However, the manuscript does not present sufficient novelty or improvements. It is also not very clear about the system and transmitted signal models and does not comprehensively evaluate the performance of the proposed methods in comparison with previous studies. Sadly, I cannot recommend the publication of this study.

I have the following comments regarding this manuscript.

1. The system model, especially the transmitted signal model, is not clear. How is the communication data modulated? Is this a single antenna radar and communication system? The transmitted signal model is very confusing and it seems to include inconsistency. How are the WTOs and WCIs defined and why are transmitted together in a sequence? 

2. In section 2.3., what are T, and N ? And, how is B_0 determined based on the signal model given?  The communication data rate must be clearly defined and explained.

3. Sum of the autocorrelation function (SACF) is defined. However, the ambiguity function is generally used in radar studies to determine the performance of radar signals. Is SACF a kind of ambiguity function here?

4. Since the system and signal models are not clear, it is not possible to evaluate the proposed JRC‐CWG and  DR‐JRC‐CWG methods. Therefore, the authors must clearly explain the system and transmitted signal models and how the communication data is transmitted.

5. In Fig. 3., PSLR is given as a function of the number of waveforms. What does 'number of waveforms' mean? Why does transmitting more waveforms decrease PSLR?

6. In Fig.4., PSLR increases as the number of WCI increases. Why does this happen? 

7. Fig.7. presents BER of JRC waveforms and BPSK. Firstly, BER is a very accurate performance metric in this study as it does not give any information regarding the data rate. Communication throughput (data rate) would be a much better performance metric to evaluate communication performance. What is the data rate offered by the different methods discussed here?

8. Moreover, BPSK has a very low spectral efficiency and it is not generally used by modern communication systems for high data communication. Therefore, it is important to provide an evaluation of higher-order PSK or QAM modulation techniques.

9. The performance of the radar must also be evaluated carefully. For example, the authors can provide Cramer-Rao lower bounds on the parameter estimation or received radar SNR to evaluate the radar metrics.

10. It is important to provide if there is a trade-off between the communication data rate and radar sensing performance in the proposed system. 

11. The proposed methods need to be compared with other dual-function waveforms in terms of data rate and radar performance to evaluate the significance of the proposed techniques.

Reviewer 2 Report

In this work, two JRC waveforms design methods are proposed. One is called the JRC complementary waveform group (JRCCWG), which addresses high range sidelobe, and the other is called the Doppler resilient JRC complementary waveform group (DRJRCCWG), which is aimed to improve the Doppler resilience of the JRCCWG. The theory is well described and detailed comparisons with simulations are provided.

While the work is interesting, the author has following concerns:

1: Throughout the paper, there are many irregularities in writing, such as line spacing.

2: As an important indicator for evaluating the performance of integrated waveforms, how is spectrum utilization evaluated?

3: In Figure 4, the PSLR increases substantially as Mc changes from 8 to 9, how can this be explained?

4: From Figure 5(b2) and Figure 5 (c2), it can be seen that the Doppler resilience of the designed DRJRCCWG will be improved when the weight w1 is increased, and it can be seen from Figure 6 that when the fd is larger, PSLR also improves with the increase of the weight w1, so is the larger w1, the better the two performances?